# Comparative Study of Short-Term Efficacy and Safety of Mitomycin versus Lobaplatin for Hyperthermic Intraperitoneal Chemotherapy after Radical Surgery in Colorectal Cancer with High-Risk Factors for Peritoneal Carcinomatosis: A Propensity Score Matching Analysis

**Xikai Guo** [†][ID]**, Yao Lin** [†]**, Chu Shen, Yuan Li, Fan Xiang, Tuo Ruan, Xinyu Zeng, Jianbo Lv, Kaixiong Tao** * **and Chuanqing Wu** *

Department of Gastrointestinal Surgery, Union Hospital, Tongji Medical College, Huazhong University of Science and Technology, Wuhan 430022, China

* Correspondence: kaixiongtao@hust.edu.cn (K.T.); wucq2014@hust.edu.cn (C.W.); Tel.: +86-027-85351662 (K.T.); Tel.: +86-027-85351620 (C.W.)

† These authors contributed equally to this work.

**Abstract:** Background: The drug selection of radical surgery (RS), with hyperthermic intraperitoneal chemotherapy (HIPEC), in pT4 colorectal cancer (CRC) remains controversial. Methods: Adverse events after HIPEC were estimated by common terminology criteria for adverse events version 5.0. The efficacy was evaluated using overall survival (OS) and recurrence-free rate (RFR). Propensity score matching (PSM) was used to reduce the influence of confounders between Mitomycin and Lobaplatin groups. Results: Of the 146 patients, from April 2020 to March 2021, 47 were managed with mitomycin and 99 with lobaplatin. There was no significant difference in the incidence of all adverse events between the two groups after PSM. OS and RFR were not significantly different between the two groups at 22 months ($p = 0.410$; $p = 0.310$). OS and RFR of the two groups also showed no significant difference for patients with T4a or T4b stage, tumor size < or ≥ 5 cm. Among patients with colon cancer, RFR at 22 months of the two groups was significantly different (100.0% vs. 63.2%, $p = 0.028$). Conclusions: In summary, the safety of mitomycin and lobaplatin for HIPEC was not different. Compared with lobaplatin, mitomycin for HIPEC after RS could benefit patients with colon cancer in RFR.

**Keywords:** hyperthermic intraperitoneal chemotherapy; colorectal cancer; mitomycin; lobaplatin; propensity score matching



## 1. Introduction

At present, the proportion of early diagnoses of colorectal cancer (CRC) in China is only 5–10%. Most CRC are found in the advanced stage or even end-stage [1]. The prognosis of CRC with peritoneal metastasis is poor, and the median OS of its natural course is only 5–7 months [2]. There are several reports confirming that hyperthermic intraperitoneal chemotherapy (HIPEC) can clear and kill free cancer cells in the peritoneal cavity, which can be used as an effective adjuvant treatment for preventing peritoneal metastasis after surgery of advanced peritoneal tumor [3–7]. There were studies [8–12] that reported that advanced stage, tumor perforation, completeness of cytoreduction, histologic type, tumor location, and nodal stage associated with peritoneal metastasis from CRC. Patients with T4 CRC have a 3-year disease-free survival (DFS) rate of only 58% to 72% after curative surgery and regular adjuvant therapy [13,14]. Meanwhile, studies have shown that the rate of peritoneal metastasis is as high as 15% to 36.7% at 3 years after surgery in

T4 CRC [10]. How to effectively prevent the recurrence and metastasis of T4 CRC is the hotspot and difficult point of current research.

To improve patients' survival and quality of life, positive intraperitoneal treatments, including radical CRC surgery and HIPEC, have been introduced in many reports [15]. However, according to some clinical studies, whether adjuvant HIPEC in T4 CRC improves overall survival and peritoneal metastasis-free survival is still controversial [16,17]. For T4 CRC, there are no clinical guidelines that provide standardized drug treatment procedures for adjuvant HIPEC. HIPEC has a large variety of drugs, and issues such as dose, timing, and drug combinations are inconclusive so far. The standardized clinical application of HIPEC medication becomes an urgent critical issue. [18] Thus, it is important to investigate the safety and efficacy of different chemotherapeutic agents used for HIPEC treatment of CRC, in order to improve the survival rate and the quality of survival of patients with advanced CRC.

Therefore, this retrospective cohort study aimed to investigate the safety and efficacy of drug selection for tumor radical surgery combined with adjuvant HIPEC in patients with T4 CRC. To provide a safety and feasibility reference for future large-scale development of treatment and randomized controlled trials (RCT).

## 2. Patients and Methods

### 2.1. Study Cohort

We reviewed the database of phase II prospective RCT (NCT04845490). Hospital records of patients diagnosed with T4 CRC treated with radical surgery (RS) combined with HIPEC from April 2020 to March 2021 in Union Hospital, Tongji Medical College, Huazhong University of Science and Technology were obtained from the RCT database. To be included, study subjects were supposed to satisfy the following criteria: (1) It was confirmed by postoperative pathology as advanced colorectal adenocarcinoma ($pT \geq 4$), (2) No distant metastasis was found before the operation, (3) Drugs used for HIPEC were mitomycin or lobaplatin and (4) Patients with rectal carcinoma is above the peritoneal reflection. MRI and colonoscopy before the operation showed that the distance from the lower edge of the tumor to the anal margin was at least 7cm (above the peritoneal reflection). Exclusion criteria were designed as followed: (1) Patients with incomplete clinical and pathological data, (2) Tumor metastasis was found during operation, (3) Patients had a history of other malignancies within 5 years, (4) emergency surgery due to perforation, ileus, etc. This retrospective study was approved by our institutional review board and the patients' information and privacy should be closely protected. Institutional ethical approval (2020) LUNSHENGZI (0450) was received for the study with informed consent. All study subjects gave full and informed consent to participate in the retrospective study.

The patients were divided into two groups: the mitomycin group and the lobaplatin group according to the drug they received during HIPEC treatment.

### 2.2. Data Collection

The data of patients were collected through the electronic case database system of the hospital, including gender, age, height, weight, past history, American Society of Anesthesiologists (ASA) classification, operation method, tumor location, maximum tumor diameter, tumor histological types, surgical margin, nerve invasion, vascular invasion, lymphatic metastasis, depth of invasion, the length of postoperative hospital stay and HIPEC information. Body mass index (BMI); Charlson comorbidity index (CCI) was used to evaluate the preoperative basic medical history; lymph node metastasis and depth of invasion were evaluated by the American Joint Committee on Cancer (AJCC) TNM staging system.

The common terminology criteria for adverse events (CTCAE 5.0) were used to record the grade 2 and above adverse events which occurred after the operation with HIPEC: Anemia, hypoalbuminemia, wound complications, anastomotic leakage, myelo-

suppression, abdominal infection, pulmonary infection, postoperative bleeding, intestinal obstruction, electrolyte disorder (Supplementary Table S1).

### 2.3. Follow-Up

Follow-up data were collected from the outpatients' visits. Patients were followed up every three months during the first two years after surgery. Routine blood tests, liver and kidney function, tumor markers, chest, abdominal and pelvic CT examination should be performed in outpatient reexamination. According to the specific situation of patients, MRI, gastroscopy, endoscopy, and other examinations were chosen.

The evaluation of effectiveness includes the following aspects: (1) tumor efficacy indicators include tumor recurrence (location and time), metastasis (location and time), and death (cause and time), (2) Overall survival (OS): refers to the time from the beginning of initial diagnosis to death due to any cause or the end of follow-up. (3) Recurrence-free rate (RFR): the time from RS until the patient's recurrence (local and peritoneal recurrence) or last contact and the time point of progression was the date of the first detectable new lesions.

### 2.4. Treatment

None of the patients included in this study received neoadjuvant therapy or radiotherapy. All patients had been informed about the operation procedure and HIPEC therapy due to their cT4 preoperative staging and voluntarily signed an informed consent form for the procedure. The HIPEC was implemented immediately after RS. Adjuvant HIPEC was conducted according to the formal standards and specifications of the clinical application of HIPEC in China, constituted by the Peritoneal Surface Oncology Committee of the China Anti-Cancer Association [19]. Clinicians adopted the open or laparoscopic surgical approach and performed radical resection of CRC (resection of the corresponding colon and rectum plus regional lymph node dissection, including the paraintestinal, middle, and mesorectal root lymph nodes), following the principles of mesorectal excision and the principles of tumor free operation, and the surgical code was referred to Chinese protocol of diagnosis and treatment of CRC (2020 edition) [1]. Intraoperatively, two perfusion tubes (inflow tube) and two drainage tubes (outflow tube) were placed through the abdominal wall. The extracorporeal circulation pipes were connected to the HIPEC machine (BRTRG-I, Bright Medical Tech, Guangzhou, China) (Figure 1). The perfusion flow rate was 400 mL/min, the perfusion time was 90 min for mitomycin ($35 \text{ mg/m}^2$) and 60 min for lobaplatin ($35 \text{ mg/m}^2$), the perfusion bag was filled with normal saline $2\text{L/m}^2$, and the perfusion temperature was $(43 \pm 0.5)°C$. Postoperative adjuvant chemotherapy was started 3–4 weeks after HIPEC, and patients with poor physical condition were appropriately extended, but at the latest, it should not exceed 8 weeks after surgery. The chemotherapy regimen was decided by clinicians according to the pathological stage, molecular classification, risk factors, referring to NCCN guidelines (2018) [20]. XELOX or mFOLFOX6 regimen was preferentially recommended.

### 2.5. Statistical Analysis

IBM SPSS Statistics v26.0 (IBM, Northridge, CA, USA) software was used for statistical processing and GraphPad prism v8.0 software was used for mapping. Classification data were expressed as percentages, continuous data as mean or median, and survival data as Kaplan–Meier curves. The measurement data in accordance with the normal distribution were expressed as mean ±SD and Student's *t*-test was used for comparison between groups; the counting data were expressed as (%) and the comparison between the groups was performed by the $\chi^2$ test. Propensity score matching (PSM) was used to reduce the influence of these biases and confounding variables so as to make a reasonable comparison between groups. PSM was conducted on the basis of 15 clinically relevant variables in Table 1. The quality of the matching was assessed by absolute standard differences, with a value <5% considered as not significant. Bilateral $p < 0.05$ means the difference is statistically significant.

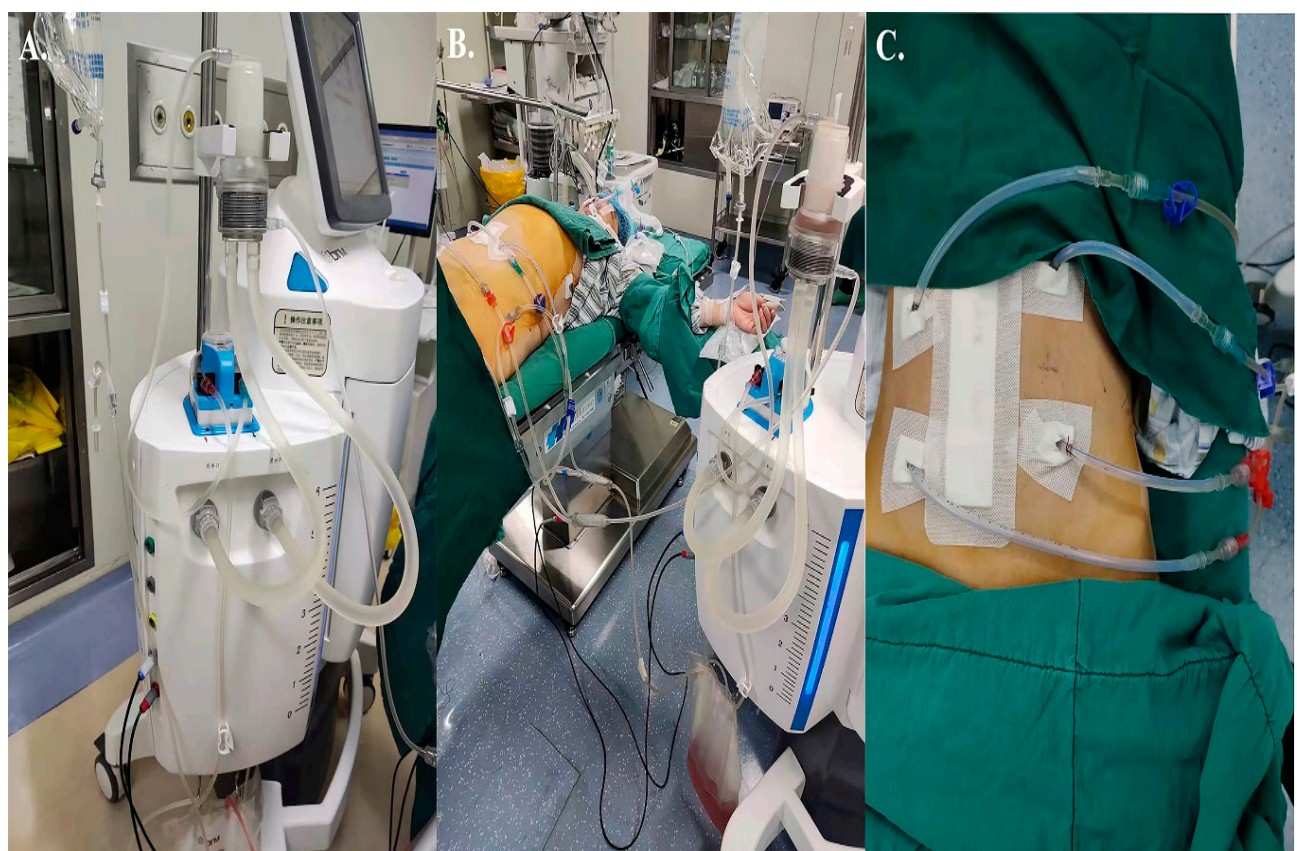

**Figure 1.** Perfusion equipment and pipe connection. (**A**) The BRTRG-I hyperthermic perfusion intraperitoneal treatment system (Bright Medical Tech, Guangzhou, China). (**B**) Pipe connection. The red tube is the inlet tube, and the blue tube is the outlet tube. The perfusion fluid enters the abdominal cavity from the red tube below and flows out through the blue tube above. (**C**) Four tubes were used for circulatory perfusion; two each were placed in the upper and lower abdomen.

**Table 1.** Baseline clinicopathological characteristics of patients with T4 colorectal cancer before PSM.

| Characteristic | Overall (*n* = 146) | No (%) Mitomycin (*n* = 47) | Lobaplatin (*n* = 99) | $\chi^2$/Z | *p* Value |
|---|---|---|---|---|---|
| Age, years | | | | −2.067 | 0.040 |
| Mean ± SD | 57.79 ± 11.092 | 55.06 ± 9.841 | 59.08 ± 11.460 | | |
| ≤60 | 81 (55.5) | 30 (63.8) | 51 (51.5) | | |
| >60 | 65 (0.445) | 17 (36.2) | 48 (48.5) | | |
| Sex | | | | 1.706 | 0.191 |
| Female | 61 (41.8) | 16 (34.0) | 45 (45.5) | | |
| Male | 85 (58.2) | 31 (66.0) | 54 (54.5) | | |
| BMI | | | | 0.402 | 0.689 |
| Mean ± SD | 23.50 ± 3.690 | 23.68 ± 4.296 | 23.42 ± 3.386 | | |
| Charlson Comorbidity Index | | | | 0.282 | 0.963 |
| 0 | 108 (74.0) | 34 (72.3) | 74 (74.7) | | |
| 1 | 28 (19.2) | 10 (21.3) | 18 (18.2) | | |
| 2 | 6 (4.1) | 2 (4.3) | 4 (4.0) | | |
| ≥3 | 4 (2.7) | 1 (2.1) | 3 (3.0) | | |
| ASA Score | | | | 6.974 | 0.031 |
| 1 | 31 (21.2) | 15 (31.9) | 16 (16.2) | | |
| 2 | 109 (74.7) | 32 (68.1) | 77 (77.8) | | |
| ≥3 | 6 (4.1) | 0 (0.0) | 6 (6.1) | | |

**Table 1.** *Cont.*

| Characteristic | Overall (*n* = 146) | No (%) Mitomycin (*n* = 47) | Lobaplatin (*n* = 99) | $\chi^2$/Z | *p* Value |
|---|---|---|---|---|---|
| Surgical Procedures | | | | 0.002 | 0.966 |
| laparoscopy | 143 (97.9) | 46 (97.9) | 97 (98.0) | | |
| laparotomy | 3 (2.1) | 1 (2.1) | 2 (2.0) | | |
| Tumor Location | | | | 6.372 | 0.041 |
| right semicolon | 43 (29.5) | 14 (29.8) | 29 (29.3) | | |
| left semicolon | 50 (34.2) | 22 (46.8) | 28 (28.3) | | |
| rectum | 53 (36.3) | 11 (23.4) | 42 (42.4) | | |
| Tumor Size, cm | | | | | |
| Mean ± SD | 4.848 ± 2.320 | 4.681 ± 1.944 | 4.927 ± 2.484 | −0.598 | 0.551 |
| ≤5 | 104 (71.2) | 38 (80.9) | 66 (66.7) | | |
| >5 | 42 (28.8) | 9 (19.1) | 33 (33.3) | | |
| Tumor differentiation | | | | 1.900 | 0.168 |
| poor or undifferentiation | 12 (8.2) | 6 (12.8) | 6 (6.1) | | |
| Well or moderately | 134 (91.8) | 41 (87.2) | 93 (93.9) | | |
| pT status | | | | 0.067 | 0.796 |
| pT4a | 91 (62.3) | 30 (63.8) | 61 (61.6) | | |
| pT4b | 55 (37.7) | 17 (36.2) | 38 (38.4) | | |
| No. of resected lymph nodes | | | | 2.018 | 0.045 |
| Mean ± SD | 21.39 ± 7.599 | 23.21 ± 8.382 | 20.53 ± 7.079 | | |
| pN status | | | | 8.920 | 0.063 |
| pN0 | 83 (56.8) | 29 (61.7) | 54 (54.5) | | |
| pN1a | 21 (14.4) | 11 (23.4) | 10 (10.1) | | |
| pN1b | 25 (17.1) | 4 (8.5) | 21 (21.2) | | |
| pN2a | 7 (4.8) | 1 (2.1) | 6 (6.1) | | |
| pN2b | 10 (6.8) | 2 (4.3) | 8 (8.1) | | |
| nerve invasion | | | | 0.934 | 0.334 |
| No | 86 (58.9) | 25 (53.2) | 61 (61.6) | | |
| Yes | 60 (41.1) | 22 (46.8) | 38 (38.4) | | |
| vascular invasion | | | | 0.126 | 0.723 |
| No | 90 (61.6) | 28 (59.6) | 62 (62.6) | | |
| Yes | 56 (38.4) | 19 (40.4) | 37 (37.4) | | |
| MMR positive | | | | 0.109 | 0.741 |
| No | 138 (94.5) | 44 (93.6) | 94 (94.9) | | |
| Yes | 8 (5.5) | 3 (6.4) | 5 (5.1) | | |
| Adjuvant chemotherapy | | | | 0.855 | 0.355 |
| XELOX | 98 (67.1) | 34 (72.3) | 64 (64.6) | | |
| mFOLFOX6 | 48 (32.9) | 13 (27.7) | 35 (35.3) | | |
| Follow-up time, months | | | | −2.439 | 0.016 |
| Mean ± SD | 15.36 ± 2.713 | 14.57 ± 2.320 | 15.73 ± 2.817 | | |
| Median(range) | 15(12–22) | 13(12–20) | 15(12–22) | | |

PSM: propensity score matching; SD:standard deviation; BMI: body mass index; ASA: American society of anesthesiologists; MMR: mismatch repair mutation.

## 3. Results

### 3.1. Patients' Characteristics

According to the inclusion and exclusion criteria, we identified a total of 146 patients, 47 managed with Mitomycin and 99 with Lobaplatin (Figure 2). There were 85 men and 61 women with a median age of 58 years. After PSM, there were 43 men and 19 women with a median age of 57.5 years. The remaining baseline clinicopathological characteristics of patients are listed in Table 1. There was no significant difference in statistics except for age ($p$ = 0.040, $\chi^2$ = −2.067), ASA score ($p$= 0.031, $\chi^2$ = 6.974), tumor location ($p$ = 0.041, $\chi^2$ = 6.372) and the number of resected lymph nodes ($p$ = 0.045, $\chi^2$ = 2.018) between Mitomycin group and Lobaplatin group.

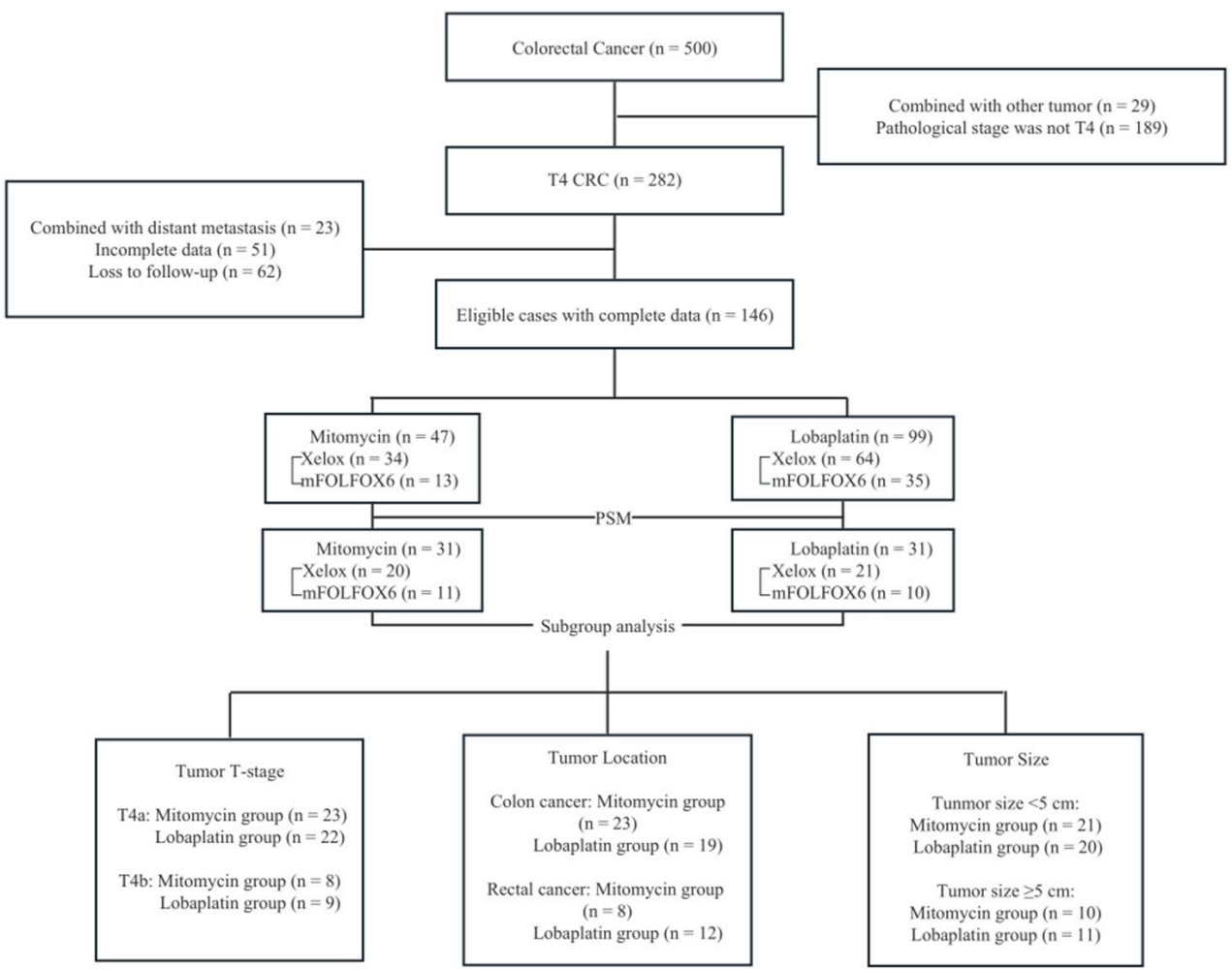

**Figure 2.** Flow chart. A total of 146 out of 500 T4 CRC patients were included in the study. Inclusion criteria: (1) It was confirmed by postoperative pathology as advanced colorectal adenocarcinoma (pT ≥ 4), (2) No distant metastasis was found before the operation, (3) Drugs used for HIPEC were mitomycin or lobaplatin and (4) Patients with rectal carcinoma is above the peritoneal reflection. MRI or colonoscopy before the operation showed that the distance from the lower edge of the tumor to the anal margin was at least 7cm (above the peritoneal reflection). Exclusion criteria were designed as followed: (1) Patients with incomplete clinical and pathological data, (2) Tumor metastasis was found during operation, (3) Patients had a history of other malignancies within 5 years, (4) emergency surgery due to perforation, ileus, etc.

After PSM, there was no significant difference in statistics for all baseline clinico-pathological characteristics between the Mitomycin group and the Lobaplatin group. All *p* values > 0.05. Baseline clinicopathological characteristics of patients after PSM are listed in Table 2.

**Table 2.** Baseline clinicopathological characteristics of patients with T4 colorectal cancer after PSM.

| Characteristic | Overall (*n* = 62) | No (%) Mitomycin (*n* = 31) | Lobaplatin (*n* = 31) | $\chi^2$/Z | *p* Value |
|---|---|---|---|---|---|
| Age, years | | | | | |
| Mean ± SD | 59.02 ± 11.810 | 56.29 ± 8.844 | 57.90 ± 11.825 | −0.608 | 0.545 |
| ≤60 | 36 (58.1) | 19 (61.3) | 17 (54.8) | | |
| >60 | 26 (41.9) | 12 (38.7) | 14 (45.2) | | |

**Table 2.** *Cont.*

| Characteristic | Overall (*n* = 62) | No (%) Mitomycin (*n* = 31) | Lobaplatin (*n* = 31) | $\chi^2$/Z | *p* Value |
|---|---|---|---|---|---|
| Sex | | | | 0.076 | 0.783 |
| Female | 19 (30.6) | 9 (29.0) | 10 (32.3) | | |
| Male | 43 (69.4) | 22 (71.0) | 21 (67.7) | | |
| BMI | | | | | |
| Mean ± SD | 22.840 ± 3.369 | 23.899 ± 4.830 | 23.950 ± 3.561 | −0.047 | 0.962 |
| Charlson Comorbidity Index | | | | 0.828 | 0.843 |
| 0 | 43 (69.4) | 20 (64.5) | 23 (74.2) | | |
| 1 | 14 (22.6) | 8 (25.8) | 6 (19.4) | | |
| 2 | 3 (4.8) | 2 (6.5) | 1 (3.2) | | |
| ≥3 | 2 (3.2) | 1 (3.2) | 1 (3.2) | | |
| ASA Score | | | | 1.160 | 0.560 |
| 1 | 13 (21.0) | 6 (19.4) | 7 (22.6) | | |
| 2 | 48 (77.4) | 25 (80.6) | 23 (74.2) | | |
| 3 | 1 (1.6) | 0 (0.0) | 1 (3.2) | | |
| Surgical Procedures | | | | <0.001 | 1.000 |
| laparoscopy | 60 (96.8) | 30 (96.8) | 30 (96.8) | | |
| laparotomy | 2 (3.2) | 1 (3.2) | 1 (3.2) | | |
| Tumor Location | | | | 1.182 | 0.554 |
| right semicolon | 20 (32.3) | 11 (35.5) | 9 (29.0) | | |
| left semicolon | 22 (35.5) | 12 (38.7) | 10 (32.3) | | |
| rectum | 20 (32.3) | 8 (25.8) | 12 (38.7) | | |
| Tumor Size, cm | | | | | |
| Mean ± SD | 4.73 ± 2.127 | 4.58 ± 1.852 | 4.84 ± 3.081 | −0.405 | 0.687 |
| ≤5 | 47 (75.8) | 25 (80.6) | 22 (71.0) | | |
| >5 | 15 (24.2) | 6 (19.4) | 9 (29.0) | | |
| Tumor differentiation | | | | 2.296 | 0.130 |
| poor or undifferentiation | 8 (12.9) | 2 (6.5) | 6 (19.4) | | |
| Well or moderately | 54 (87.1) | 29 (93.5) | 25 (80.6) | | |
| pT status | | | | 0.081 | 0.776 |
| pT4a | 45 (72.6) | 23 (74.2) | 22 (71.0) | | |
| pT4b | 17 (27.4) | 8 (25.8) | 9 (29.0) | | |
| No. of resected lymph nodes | | | | 1.557 | 0.125 |
| Mean ± SD | 21.15 ± 9.416 | 23.06 ± 8.342 | 20.23 ± 5.783 | | |
| pN status | | | | 2.816 | 0.589 |
| pN0 | 40 (64.5) | 22 (71.0) | 18 (58.1) | | |
| pN1a | 7 (11.3) | 4 (12.9) | 3 (9.7) | | |
| pN1b | 11 (17.7) | 3 (9.7) | 8 (25.8) | | |
| pN2a | 2 (3.2) | 1 (3.2) | 1 (3.2) | | |
| pN2b | 2 (3.2) | 1 (3.2) | 1 (3.2) | | |
| nerve invasion | | | | 0.265 | 0.607 |
| No | 36 (58.1) | 17 (54.8) | 19 (61.3) | | |
| Yes | 26 (41.9) | 14 (45.2) | 12 (38.7) | | |
| vascular invasion | | | | <0.001 | 1.000 |
| No | 40 (64.5) | 20 (64.5) | 20 (64.5) | | |
| Yes | 22 (35.5) | 11 (35.5) | 11 (35.5) | | |
| MMR positive | | | | 0.218 | 0.641 |
| No | 57 (91.9) | 28 (90.3) | 29 (93.5) | | |
| Yes | 5 (8.1) | 3 (9.7) | 2 (6.5) | | |
| Adjuvant chemotherapy | | | | 0.072 | 0.788 |
| XELOX | 41 (66.1) | 20 (64.5) | 21 (67.7) | | |
| mFOLFOX6 | 21 (33.9) | 11 (35.5) | 10 (32.3) | | |
| Follow-up time, months | | | | −0.943 | 0.349 |
| Mean ±SD | 18.95 ± 3.857 | 15.13± 2.487 | 15.77 ± 2.883 | | |
| Median(range) | 15.5 (12–22) | 16 (12–20) | 15 (12–22) | | |

PSM: propensity score matching; SD:standard deviation; BMI: body mass index; ASA: American society of anesthesiologists; MMR: mismatch repair mutation.

### 3.2. Adverse Events

Before PSM, there was no significant difference in statistics for anemia ($p = 0.336$, $\chi^2 = 0.927$), hypoalbuminemia ($p = 0.809$, $\chi^2 = 0.059$), myelosuppression ($p = 1.000$, $\chi^2 < 0.001$), wound complications ($p = 0.082$, $\chi^2 = 2.971$), abdomen infection ($p = 0.145$, $\chi^2 = 2.121$), pulmonary infection ($p = 0.755$, $\chi^2 = 0.097$), postoperative bleeding ($p = 0.587$, $\chi^2 = 0.295$), anastomotic leakage ($p = 0.327$, $\chi^2 = 0.963$), ileus ($p = 0.145$, $\chi^2 = 2.121$), liver dysfunction ($p < 0.001$, $\chi^2 = 1.000$), electrolyte disorder($p = 0.440$, $\chi^2 = 0.598$) and digestive symptoms ($p = 0.162$, $\chi^2 = 1.952$) between Mitomycin group and Lobaplatin group.

After PSM, there was no significant difference in statistics for all adverse events between the Mitomycin group and the Lobaplatin group. Adverse events before and after PSM are listed in Table 3.

**Table 3.** Adverse events of patients with advanced colorectal cancer before and after PSM.

| Adverse Event | Before PSM | | | | | After PSM | | | | |
|---|---|---|---|---|---|---|---|---|---|---|
| | No (%) | | | $\chi^2$ | *p* Value | No (%) | | | $\chi^2$ | *p* Value |
| | Overall ($n = 146$) | Mitomycin ($n = 47$) | Lobaplatin ($n = 99$) | | | Overall ($n = 62$) | Mitomycin ($n = 31$) | Lobaplatin ($n = 31$) | | |
| Anemia | 25 (17.1) | 6 (12.8) | 19 (19.2) | 0.927 | 0.336 | 13 (21.0) | 5 (16.1) | 8 (25.8) | 0.876 | 0.349 |
| Hypoalbuminemia | 36 (24.7) | 11 (23.4) | 25 (25.3) | 0.059 | 0.809 | 17 (27.4) | 6 (19.4) | 11 (35.5) | 2.026 | 0.155 |
| Myelosuppression | 0 (0.0) | 0 (0.0) | 0 (0.0) | 0.000 | 1.000 | 0 (0.0) | 0 (0.0) | 0 (0.0) | 0.000 | 1.000 |
| Wound complications | 6 (4.1) | 0 (0.0) | 6 (6.1) | 2.971 | 0.082 | 1 (1.6) | 0 (0.0) | 1 (3.2) | 1.016 | 0.313 |
| Abdomen infection | 1 (0.7) | 1 (2.1) | 0 (0.0) | 2.121 | 0.145 | 0 (0.0) | 0 (0.0) | 0 (0.0) | 0.000 | 1.000 |
| Pulmonary infection | 4 (2.7) | 1 (2.1) | 3 (3.0) | 0.097 | 0.755 | 2 (3.2) | 1 (3.2) | 1 (3.2) | 0.000 | 1.000 |
| Postoperative bleeding | 2 (1.4) | 1 (2.1) | 1 (1.0) | 0.295 | 0.587 | 2 (3.2) | 1 (3.2) | 1 (3.2) | 0.000 | 1.000 |
| Anastomotic leakage | 2 (1.4) | 0 (0.0) | 2 (2.0) | 0.963 | 0.327 | 1 (1.6) | 0 (0.0) | 1 (3.2) | 1.016 | 0.313 |
| Ileus | 1 (0.7) | 1 (2.1) | 0 (0.0) | 2.121 | 0.145 | 1 (1.6) | 1 (3.2) | 0 (0.0) | 1.016 | 0.313 |
| Liver dysfunction | 0 (0.0) | 0 (0.0) | 0 (0.0) | 0.000 | 1.000 | 0 (0.0) | 0 (0.0) | 0 (0.0) | 0.000 | 1.000 |
| Electrolyte disturbance | 4 (2.7) | 2 (4.3) | 2 (2.0) | 0.598 | 0.440 | 2 (3.2) | 1 (3.2) | 1 (3.2) | 0.000 | 1.000 |
| Digestive symptoms | 4 (2.7) | 0 (0.0) | 4 (4.0) | 1.952 | 0.162 | 1 (1.6) | 0 (0.0) | 1 (3.2) | 1.016 | 0.313 |

PSM: propensity score matching.

### 3.3. Comparison of Short-Term Prognosis

Before PSM, survival analysis revealed that patients receiving HIPEC with Lobaplatin combined with RS had a lower cumulative OS than those receiving Mitomycin, with 99.0% vs. 100.0% at 22 months, respectively. Also, the Lobaplatin group had a lower cumulative RFR than the Mitomycin group with 80.2% vs. 95.7% at 22 months. But in the corresponding cumulative OS and RFR K-M survival curve had no significant difference through the Log-Rank test ($p = 0.410$, $\chi^2 = 0.678$; $p = 0.310$, $\chi^2 = 1.032$) (Figure 3).

After PSM, the Lobaplatin group had a lower cumulative RFR than the Mitomycin group with 75.3% vs. 96.8% at 22 months. As well, Mitomycin group had an equal cumulative OS to the Lobaplatin group with 100.0% at 22 months. However, according to the K-M survival curve, the OS and RFR of the two groups had no significant difference ($p = 1.000$, $\chi^2 < 0.001$; $p = 0.194$, $\chi^2 = 1.688$). (Figure 4)

### 3.4. Stratified Analysis of Prognosis

Stratified analysis was used to further explore the influencing factors of OS and RFR in CRC. After PSM, for stage T4a tumors ($n = 45$), the OS and RFR of both groups ($n_{Mitomycin} = 23$; $n_{Lobaplatin} = 22$) showed no significant difference ($p = 1.000$, $\chi^2 < 0.001$ and $p = 0.517$, $\chi^2 = 0.420$, respectively). There were also no significant differences ($p = 1.000$, $\chi^2 < 0.001$ and $p = 0.242$, $\chi^2 = 1.371$, respectively) in the OS and RFR of both groups ($n_{Mitomycin} = 8$; $n_{Lobaplatin} = 9$) in T4b stage tumors ($n = 17$) on the Kaplan–Meier survival curve.

After PSM, for tumor size < 5 cm ($n$ = 41), the OS and RFR of the two groups ($n_{Mitomycin}$ = 21; $n_{Lobaplatin}$ = 20) showed no significant differences ($p$ = 1.000, $\chi^2$ < 0.001 and $p$ = 0.261, $\chi^2$ = 1.262, respectively). In addition, for tumor size ≥5 cm ($n$ = 21), the OS and RFR of both groups ($n_{Mitomycin}$ = 10; $n_{Lobaplatin}$ = 11) were not significantly different ($p$ = 1.000, $\chi^2$ < 0.001 and $p$ = 0.617, $\chi^2$ = 0.250, respectively).

After PSM, with respect to rectal cancer ($n$ = 20), the OS and RFR of both groups ($n_{Mitomycin}$ = 8; $n_{Lobaplatin}$ = 12) showed no significant difference ($p$ = 1.000, $\chi^2$ < 0.001 and $p$ = 0.221, $\chi^2$ = 1.500, respectively). For colon cancer ($n$ = 42), the Mitomycin group ($n$ = 23) had better cumulative RFR (100.0% vs. 63.2%) at 22 months than those in the Lobaplatin group ($n$ = 19) ($p$ = 0.028, $\chi^2$ = 4.824, the Hazard Ratio: 0.1098, 95% CI of ratio: 0.0153–0.7884) (Figure 5). However, the OS of both groups was not significantly different ($p$ = 1.000, $\chi^2$ < 0.001).

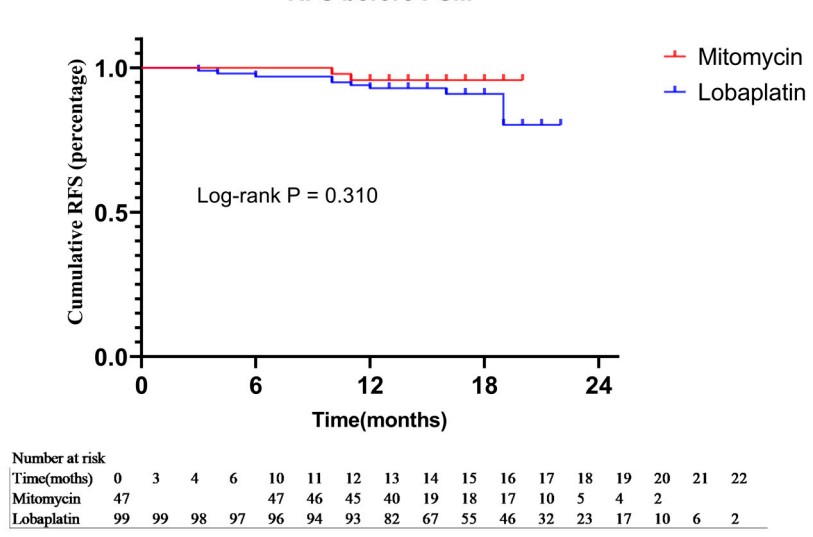

**Figure 3.** Kaplan-Meier method was used to recurrence-free survival between the Mitomycin group and Lobaplatin group before PSM.

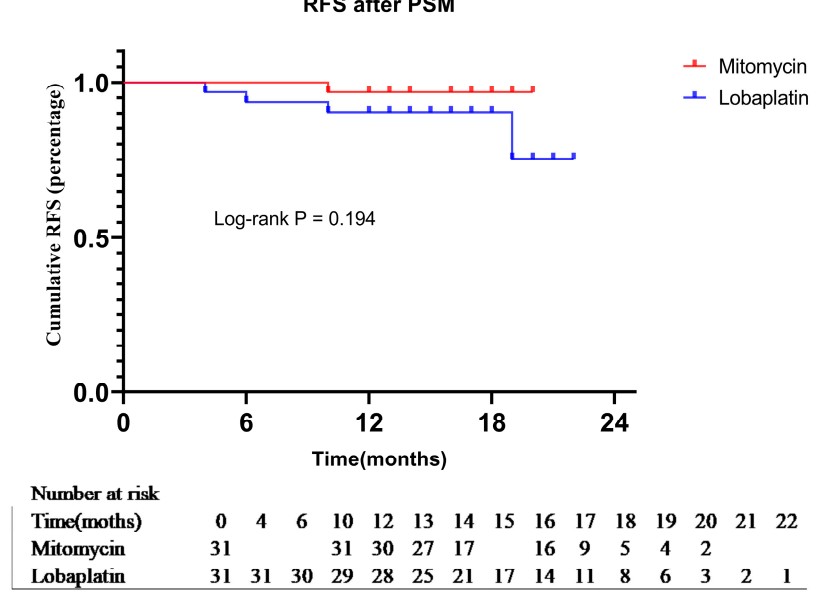

**Figure 4.** Kaplan-Meier method was used to recurrence-free survival between the Mitomycin group and Lobaplatin group after PSM.

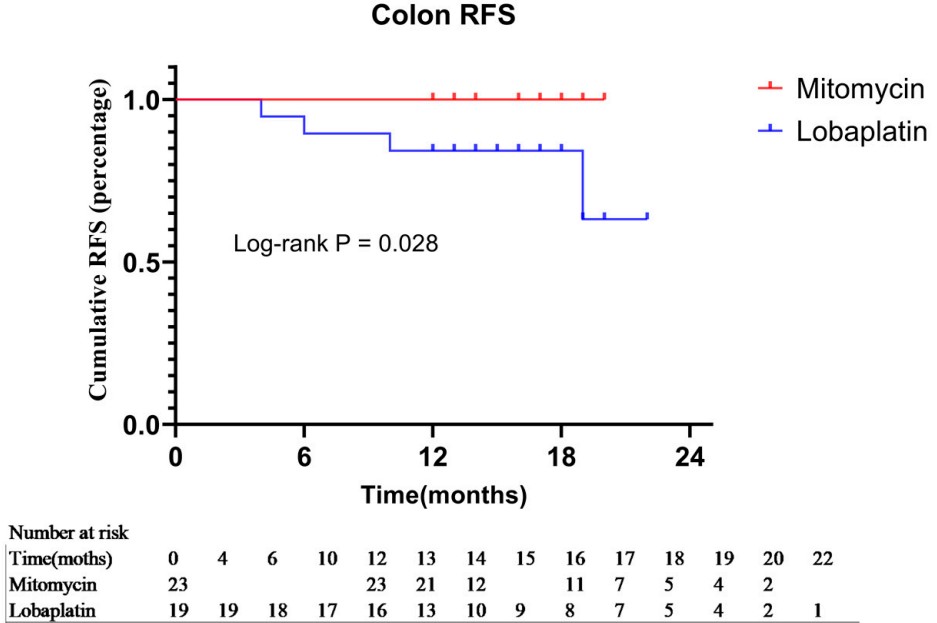

**Figure 5.** Subgroup analysis based on tumor location. Recurrence-free survival of the Mitomycin group and Lobaplatin group after PSM when the tumor at colon.

## 4. Discussion

For T4 CRC, few clinical guidelines provide standardized drug selection for adjuvant HIPEC. Drug selection for adjuvant HIPEC in CRC is inconclusive so far. Charlotte E L Klaver et al. reported that adjuvant HIPEC with oxaliplatin did not improve peritoneal metastasis-free survival at 18 months in T4 or perforated colon cancer [17]. Although the research result was negative, many scholars are still exploring the therapeutic significance of HIPEC for CRC in recent years. In the ESMO congress 2022, A. Arjona-Sanchez et al. reported that RS + HIEPC with mitomycin C for locally advanced colon cancer could improve the loco-regional control rate (97% vs. 87%, $p$ = 0.025) [21]. Similarly, our study confirmed the safety of Mitomycin or Lobaplatin for HIPEC after RS in T4 CRC and the better short-term efficacy of Mitomycin in T4 colon cancer.

Mitomycin and oxaliplatin are the most common drugs for HIPEC internationally. As early as 2003, Verwaal et al. reported that the probability of serious complications (Clavien-Dindo rating of 3/4) in the HIPEC group (fluorouracil, leucovorin) was significantly higher than that in the control group [16]. However, with the continuous improvement of HIPEC theory, technology, and equipment, the safety of HIPEC in CRC needs to be reevaluated. Klaver et al. reported that their HIPEC process was initiated by intravenous injection of fluorouracil (400 mg/m$^2$) and leucovorin (20 mg/m$^2$), then oxaliplatin (460 mg/m$^2$) was used intraperitoneally at 42 °C for 30 min. Postoperative complications occurred in 14% of patients after HIPEC, which was different from that in the control group (3%), but it was within the acceptable range [17]. Goere et al. reported that in patients receiving second-look surgery and HIPEC (oxaliplatin 460 mg/m$^2$, or oxaliplatin 300 mg/m$^2$ plus irinotecan 200 mg/m$^2$, plus intravenous fluorouracil 400 mg/m$^2$; or mitomycin 35 mg/m$^2$ alone), grade III-IV complications occurred in 41% of patients [22]. The meta-analysis shows that about 21% of patients treated with mitomycin have serious complications, while the incidence of serious complications in the oxaliplatin group is 30% [23]. Moreover, oxaliplatin is unstable in sodium chloride solution, so the commonly used perfusion solution is glucose. The study had shown that glucose as HIPEC infusion increased the risk of intraoperative hyperglycemia and postoperative infection [24]. Therefore, it may suggest that serious complications are less in mitomycin than oxaliplatin in the treatment of HIPEC. For lobaplatin, a study [25] showed that intraperitoneal perfusion chemotherapy with lobaplatin did not increase the incidence of postoperative complica-

tions and had no significant effect on bone-marrow function or liver and kidney function. There also have been some clinical studies [26,27] confirming the feasibility of lobaplatin in HIPEC. In our study, the two highest complications were 24.7% hypoalbuminemia and 17.1% anemia in the overall cohort. After removing the anemia and hypoproteinemia that often occur in postoperative patients, related complications occurred in 13% of patients after HIPEC, without a significant increase in the incidence of perioperative adverse events. There was no significant difference in statistics for all adverse events between Mitomycin and Lobaplatin groups after PSM. This study further confirmed that the perioperative safety of using mitomycin or lobaplatin for HIPEC to prevent peritoneal metastasis in patients with T4 CRC after RS was acceptable.

Since the application of HIPEC in CRC, many studies have explored the effectiveness of adjuvant HIPEC in CRC. The COLOPEC trial showed no difference in DFS at 18 months (69.0% vs. 69.3%; *p* = 0.99), OS at 18 months (93.0% vs. 94.1%; *p* = 0.82), and peritoneal metastases-free survival (PFS) at 18 months (80.9% versus 76.2%, *p* = 0.28) between the experimental and control groups. And also, in all subgroup analyses (clinical stage, pathologic stage, lymph node invasion, perforation, etc.), there was no significant difference in PFS between the two groups [17]. The PROPHYLOCHIP trial showed no difference in three-year DFS (53% vs. 44%, *p* = 0.82), five-year DFS (49% vs. 42%, *p* = 0.82), three-year PFS (61% vs. 59%), three-year OS (80% vs. 79%) and five-year OS (72% vs. 68%) between the control group and the HIPEC group after a median follow-up of 50.8 months [22,28]. The above two RCTs with oxaliplatin as the primary choice were both investigating the effect of adjuvant HIPEC in CRC with high-risk factors for peritoneal metastasis, and the current results did not yet show the effectiveness of adjuvant HIPEC. Nevertheless, the Phase III randomized HIPECT4 study [21] showed that locoregional control improved for locally advanced colon cancer in the experimental arm (35,3 ± 0.4 vs. 33.2 ± 0.8 months), with 3-year locoregional control rates of 97% and 87%, respectively (*p* = 0.025). Our study also showed that adjuvant HIPEC with mitomycin could improve the RFR of patients with T4 colon cancer at 22 months (100.0% vs. 63.2%, *p* = 0.028). But short-term OS and RFR of the two groups in other layered analyses or overall analysis did not show a significant difference before and after PSM. This may suggest that compared with platinum compounds (oxaliplatin, lobaplatin, etc.), mitomycin is more suitable for HIPEC to prevent peritoneal metastasis in T4 colon cancer after RS.

Patient selection is an important factor that influences the efficacy of HIPEC. However, few studies have reported the effect of tumor localization on the application of adjuvant HIPEC in T4 CRC. Most studies regard peritoneal metastasis of CRC as a homogeneous disease without distinguishing the origin of the colon or rectum. Ravn et al. reported that colon cancer had a higher absolute risk for metachronous peritoneal metastases 1 and 3 years after intended curative CRC surgery than rectum cancer [29]. Our study found a significant difference in RFR in advanced colon cancer, which showed that tumor localization was an important factor affecting the prognosis of T4 CRC after HIPEC + RS. Compared to lobaplatin, mitomycin prolonged RFR without adversely affecting complication rates or short-term OS. Therefore, mitomycin may be the preferential drug in RS + HIPEC procedures to prevent colonic peritoneal metastasis.

This study had some limitations. Firstly, our study was a retrospective study, whose conclusions need to be verified by prospective cohort studies. Besides, the follow-up time was not long enough to obtain a long-term prognosis. In addition, the fewer cases of recurrence and death and the shorter follow-up time may lead to an incomprehensive estimate of prognosis, which requires further long-term follow-up and more subjects of specific stratification in the later stage. Finally, we did not include the patients with T4 colorectal cancer not receiving HIPEC as a control group. It is necessary to conduct a comparative study of RS with or without HIPEC in CRC with the T4 stage. The effect of HIPEC on the long-term prognosis of CRC awaits prospective multicenter randomized controlled studies with high quality, large samples, and well-defined outcome measures.

In conclusion, the safety of adjuvant HIPEC with Mitomycin or Lobaplatin in patients with T4 CRC can be guaranteed. Moreover, regarding prognosis, patients with T4 colon cancer who underwent HIPEC with mitomycin after RS had better RFR than patients who underwent HIPEC with lobaplatin after RS. However, mitomycin did not show the expected advantages in the overall cohort or other layered analysis, which requires more subjects in the study and longer-term follow-up.

**Supplementary Materials:** The following supporting information can be downloaded at: https://www.mdpi.com/article/10.3390/curroncol30020114/s1, Table S1: Definition of CTCAE V5.0 adverse events.

**Author Contributions:** X.G.: Conceptualization, Validation, Formal analysis, Investigation, Data curation, Writing—Original Draft, Writingn Review and Editing. Y.L. (Yao Lin): Methodology, Formal analysis, Data curation, Writing—Review & Editing. C.S.: Formal analysis, Investigation. Y.L. (Yuan Li): Investigation, Resources. F.X.: and, Data curation. T.R.: Investigation, Data curation. X.Z.: Investigation. J.L.: Investigation. C.W.: Conceptualization, Resources, Supervision, Project administration, Funding acquisition. K.T.: Resources, Supervision, Project administration, Funding acquisition. All authors have read and agreed to the published version of the manuscript.

**Funding:** This research was funded by the National Nature Science Foundation of China, grant number No.82073324. Kaixiong Tao and Chuanqing Wu provided financial support for the conduct of the research and preparation of the article. Chuanqing Wu: Conceptualization, Resources, Supervision, Project administration, Funding acquisition. Kaixiong Tao: Resources, Supervision, Project administration, Funding acquisition.

**Institutional Review Board Statement:** The study was conducted in accordance with the Declaration of Helsinki, and approved by the Ethics Committee of Tongji Medical College, Huazhong University of Science and Technology ([2020] LUNSHENGZI (0450)/No: UHCT-IEC-SOP-016-02-03, approved on 10/15/2020).

**Informed Consent Statement:** The study was approved by the institutional review board at each participating site. The data are anonymous, and the requirement for informed consent was therefore waived.

**Data Availability Statement:** The original anonymous dataset is available on request from the corresponding author.

**Acknowledgments:** My sincere appreciation also goes to the teachers and students from Department of Gastrointestinal Surgery, Union Hospital, Tongji Medical College, Huazhong University of Science and Technology, who participated this study with great cooperation. Moreover, I would like to thank my girlfriend, QianYun Li, for her encouragement and support.

**Conflicts of Interest:** The authors declare no conflict of interest.

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
