# Peer review of "Comparative Study of Short-Term Efficacy and Safety of Mitomycin versus Lobaplatin for Hyperthermic Intraperitoneal Chemotherapy after Radical Surgery in Colorectal Cancer with High-Risk Factors for Peritoneal Carcinomatosis: A Propensity Score Matching Analysis"

_curroncol, doi:10.3390/curroncol30020114_

Round 1
Reviewer 1 Report
The manuscript is comprehensive, includes most of the relevant literature and discuss in depth most of the known data available for this issue. The authors described well the procedure type, drug specifications such as doses, time for administration and their dissolution. Moreover, they explained in detail the possible side effects of the the chosen treatment method among the relevant patient population. After reviewing the paper, these are my comments:
#1: The authors should clarify the mutation type of CRC if such an information is available.
# 2 In line 252 after Gore et al there is a missing full stop.
# 3 There are several papers that describe the risk factors associated with peritoneal metastasis form CRC. I added an example that the authors can use:
Yonemura Y, Canbay E, Ishibashi H. Prognostic factors of peritoneal metastases from colorectal cancer following cytoreductive surgery and perioperative chemotherapy. ScientificWorldJournal. 2013 Apr 18;2013:978394. doi: 10.1155/2013/978394. PMID: 23710154; PMCID: PMC3654240.
Author Response
Dear Reviewer,
On behalf of my co-authors, we would like to thank you for taking the time to review our manuscript. We appreciate it very much for your comments and suggestions on our manuscript entitled “Comparative study of Short-term Efficacy and Safety of Mitomycin versus Lobaplatin for Hyperthermic Intraperitoneal Chemotherapy after Radical Surgery in Colorectal Cancer with High-risk Factors for Peritoneal Carcinomatosis: A Propensity Score Matching” (Manuscript ID: curroncol-2121576). Based on the instructions provided in your comments, we uploaded the file of the original manuscript with all the changes highlighted by using the track changes mode. Thanks again!
Point 1: The authors should clarify the mutation type of CRC if such an information is available.
Response 1: We thank you very much for pointing this out. According to the guidelines(1,2) , mutation testing of RAS and BRAF genes is recommended for patients with clinically confirmed recurrent or metastatic colorectal cancer. MMR or MSI testing may be considered for all patients with CRC for Lynch syndrome screening, prognostic stratification, and to guide immunotherapy. In the research, all patients had MMR testing and we had recorded MMR mutation in our database. MMR mutation information were shown in table 1 and table 2. However, we are very sorry that we did not record RAS and BRAF mutation information since not all patients receive that testing.
[1] A. B. Benson, A. P. Venook, M. M. Al-Hawary, et al. NCCN Guidelines Insights: Colon Cancer, Version 2.2018. Journal of the National Comprehensive Cancer Network, 2018, 16(4): 359-369
[2] Hospital Authority of National Health Commission of the People's Repuhlic of China; Chinese Society of Oncology, Chinese Medical Association. Chinese protocol of diagnosis and treatment of colorectal cancer(2020 edition) Chinese Journal of Practical Surgery, 2020, 6: 601-625
Point 2: In line 252 after Gore et al there is a missing full stop.
Response 2: We are very sorry for our clerical error. Thanks for the comment and we have made the change.
Point 3: There are several papers that describe the risk factors associated with peritoneal metastasis form CRC. I added an example that the authors can use:
Yonemura Y, Canbay E, Ishibashi H. Prognostic factors of peritoneal metastases from colorectal cancer following cytoreductive surgery and perioperative chemotherapy. ScientificWorldJournal. 2013 Apr 18;2013:978394. doi: 10.1155/2013/978394. PMID: 23710154; PMCID: PMC3654240.
Response 3: Thank you for this valuable feedback. We appreciate it for your guidance. We have included the paper into Introduction (line 43-45).
We would like to express our great appreciation to you for comments on our paper. Looking forward to meet your approval.
Thank you and best regards.

Reviewer 2 Report
The authors present the outcomes from a ?retrospective single center study aiming to investigate the safety and efficacy of drug selection for tumor radical surgery combined with adjuvant HIPEC in patients with T4 CRC. This is an interesting study on what is indeed a particular group of patients on which we lack an optimal treatment plan. A number of issues to be addressed by the authors:
- How is the study retrospective when it is based on prospective RCT data?
- How many T4 patients did not get HIPEC, it would be interesting for the readership to see how their survival and recurrence was. If ALL T4 patients underwent HIPEC during the study period in this center by protocol this needs to be clearly stated. If not it should also be added in the limitations section.
- How did the authors choose to present a subgroup psm analysis on tumor size and not include N+ status? for instance.
- Supplementary figure 1 should be added as a normal figure in the methods relevant section.
Author Response
Dear Reviewer,
On behalf of my co-authors, we would like to thank you for taking the time to review our manuscript. Your invaluable and useful suggestions have enabled us to improve our work. Thanks for your positive and constructive comments on our manuscript entitled “Comparative study of Short-term Efficacy and Safety of Mitomycin versus Lobaplatin for Hyperthermic Intraperitoneal Chemotherapy after Radical Surgery in Colorectal Cancer with High-risk Factors for Peritoneal Carcinomatosis: A Propensity Score Matching” (Manuscript ID: curroncol-2121576). We have studied all comments carefully and have made correction. Revised portion are highlighted by using the track changes mode.
Point 1: How is the study retrospective when it is based on prospective RCT data?
Response 1: We thank you for raising this question. The drug selection of preventive HIPEC has not yet formed a standard scheme in the guidelines of various countries or regions. Therefore, patients would be enrolled in the clinical trials the center joined in (NCT04370925, NCT04845490), before they can get the application of mitomycin and lobplatin in HIPEC. With the consent of the hospital ethics committee (line 80), our study collected part of the data (patients using mitomycin and lobaplatin as HIPEC drugs) for retrospective analysis according to the inclusion and exclusion criteria. Not in accordance with the basic principles of RCTs: randomization, control, blinding. The prospective database was selected because ethical issues in implementing HIPEC and medication have been approved in prospective RCTs. In this study, we performed a retrospective analysis of already existing data, and did not rerandomize enrolled patients, implement extra interventions. We could look forward to the analysis of complete data from prospective RCTs (NCT 04370925, NCT 04845490).
Point 2: How many T4 patients did not get HIPEC, it would be interesting for the readership to see how their survival and recurrence was. If ALL T4 patients underwent HIPEC during the study period in this center by protocol this needs to be clearly stated. If not it should also be added in the limitations section.
Response 2: We appreciate you for this helpful recommendation and valuable suggestion, and we believe that your comments shall be very useful to our further exploration in future. Our study aimed to describe the short-term efficacy and safety of mitomycin and lobaplatin for prophylactic HIPEC after RS in patients with T4 CRC. We did not include patients not receiving HIEPC in this study. We recognize this limitation should be mentioned in the paper, so we added the following sentence to limitation in the discussion: “Finally, we did not include the patients with T4 colorectal cancer not receiving HIPEC as a control group. It is necessary to conduct comparative study of RS with or without HIPEC in CRC with T4 stage. The effect of HIPEC on long-term prognosis of CRC awaits prospec-tive multicenter randomized controlled studies with high quality, large sample, and well-defined outcome measures.” (line 320-324). We hope this was the right correction.
In China, if patients were diagnosed with cT4 colorectal cancer before surgery, we would evaluate their general condition and assess whether or not to recommend HIPEC (line 116-118). The treatment strategy was discussed preoperatively together with patients and their relatives. HIPEC was implemented after patients signed informed consent. Not all patients with T4 colorectal cancer would choose to receive HIPEC. The survival and recurrence of T4 patients who did not receive HIPEC is always our concern. HIPEC-06 prospective RCT (NCT 04370925) “A Multicenter Prospective Randomized Controlled Clinical Trial of Hyperthermic Intraperitoneal Chemotherapy after Colectomy in Patients with Colorectal Cancer at High Risk of Peritoneal Carcinomatosis” participated by our center is studying this topic. In addition, another retrospective study from our center “Comparative Study of Radical Surgery with or without Hyperthermic Intraperitoneal Chemotherapy in Colorectal Cancer with T4 stage” is under review in Journal of Clinical Medicine.
Point 3: How did the authors choose to present a subgroup psm analysis on tumor size and not include N+ status? for instance.
Response 3: We thank you for pointing this out. There are several papers 1-4 reported that advanced stage (T4), tumor perforation, tumor location and nodal stage (N2) etc. associated with peritoneal metastasis form CRC. Therefore, our study proceeded from analyzing these several high-risk factors as subgroup. We have taken into account the N stage stratification during our analysis, as shown in the figure below. But the N2 stage sample size was too small, so the stratified analysis of N stage was not put into the article. In subsequent work, we will expand the sample size and consider other stratified analysis to further explore the application value of HIPEC in specific populations. As for tumor size, we conducted subgroup analysis on it because of the following reasons: First, patients with a large tumor diameter may grow for a longer time and have a later clinical stage. Second, it may be due to relatively rapid growth, i.e., growing much larger over the same period. Patients with larger tumor volumes are more likely to experience intestinal obstruction, tumor rupture or perforation, and bleeding leading to anemia, which can all affect the prognosis of patients.
Thank you very much for your comments!
[1] Segelman J, Granath F, Holm T, Machado M, Mahteme H, Martling A. Incidence, prevalence and risk factors for peritoneal carcinomatosis from colorectal cancer. Br J Surg 2012; 99: 699–705.
[2] van Gestel YRBM, Thomassen I, Lemmens VEPP, et al. Metachronous peritoneal carcinomatosis after curative treatment of colorectal cancer. Eur J Surg Oncol 2014; 40: 963–69.
[3] Klaver CEL, van Huijgevoort NCM, de Buck van Overstraeten A, et al. Locally advanced colorectal cancer: true peritoneal tumor penetration is associated with peritoneal metastases. Ann Surg Oncol 2018; 25: 212–20.
[4] Cheynel N, Cortet M, Lepage C, Ortega-Debalon P, Faivre J, Bouvier AM. Incidence, patterns of failure, and prognosis of perforated colorectal cancers in a well-defined population. Dis Colon Rectum 2009; 52: 406–11.
Point 4: Supplementary figure 1 should be added as a normal figure in the methods relevant section.
Response 4: Thank you for the comment. We have changed supplementary figure 1 to normal figure 1.
We would like to express our great appreciation to you for comments on our paper. Looking forward to meet your approval.
Thank you and best regards.
Reviewer 3 Report
This work covers an interesting topic in peritoneal malignancies management. Evidences about HIPEC regimen are weak in general and even more in prophylactic setting. Methodology is robust having performed a propensity score analysis, even though sample size and length of followup are limited.
It is my opinion that article deserves publication and I have only a few questions for Authors:
-it is not clear if all cT4 patients received HIPEC and have been considered in analysis only patients with confirmed pT4. Selection criteria reported that pT4 are included but in treatment section (lines 113-120) Authors stated that HIPEC is performed immediately after CRC resection. At that time (unless using frozen pathology it would be impossible to have pathological confirmation). It could be interesting to analyse HIPEC side effects also in the 189 patients with unconfirmed pT4, since this will increase sample size (obviously those patients should be excluded in the OS or RFR analysis).
-survival tables in figures 2 to 4. Title should be RFS survival non RFR (rate), since curves refer to survivals (as correctly reported in figure legends), non to rates. X-axis legend has misspelling (moths).
Author Response
Dear Reviewer,
On behalf of my co-authors, we thank you very much for giving us an opportunity to revise our manuscript. We were pleased to know that our work was rated as potentially acceptable for publication in journal, subject to adequate revision. Thanks for your comments concerning our manuscript entitled “Comparative study of Short-term Efficacy and Safety of Mitomycin versus Lobaplatin for Hyperthermic Intraperitoneal Chemotherapy after Radical Surgery in Colorectal Cancer with High-risk Factors for Peritoneal Carcinomatosis: A Propensity Score Matching” (Manuscript ID: curroncol-2121576). Those comments are all valuable and helpful for revising and improving our paper. Revised portion are highlighted by using the track changes mode.
Point 1: it is not clear if all cT4 patients received HIPEC and have been considered in analysis only patients with confirmed pT4. Selection criteria reported that pT4 are included but in treatment section (lines 113-120) Authors stated that HIPEC is performed immediately after CRC resection. At that time (unless using frozen pathology it would be impossible to have pathological confirmation). It could be interesting to analyse HIPEC side effects also in the 189 patients with unconfirmed pT4, since this will increase sample size (obviously those patients should be excluded in the OS or RFR analysis).
Response 1: We appreciate you for this kind recommendation and insightful suggestion, and we believe that your comments shall be very useful to our further exploration in future. It is a pity that we did not record their detail baseline clinicopathological characteristics and adverse events, because the 189 patients with unconfirmed pT4 did not meet the inclusion criteria. Our study aimed to describe the short-term efficacy and safety of mitomycin and lobaplatin for prophylactic HIPEC after RS in patients with T4 CRC. We did not include patients with unconfirmed pT4 or not receiving HIEPC in this study. In the research, we select confirmed pT4 from cT4 containing pT4 and non-pT4 according to the inclusion criteria.
Clinically, we discuss the implementation of HIPEC with patients diagnosed with cT4 preoperatively by imaging or endoscopy (line 116-118). Not all patients with cT4 colorectal cancer would choose to receive HIPEC. HIPEC was implemented after patients signed informed consent. We would not recommend HIPEC to patients if cT3 was diagnosed preoperatively. Because the pathological report of the pathology department takes 2-3 weeks or even longer, we will not implement HIPEC until the pathological report comes out. As you comment, how to accurately determine T4 tumors before pathological samples are obtained is also a challenge to clinician. This resulted in part of patients with a clinical diagnosis of T4, in fact with a pathological diagnosis of T3, receiving HIPEC. However, D. Hompes et al. (1) reported that the incidence of peritoneal metastasis in patients with pT3 was as high as 9.3%. So it might be important to explore the application of HIPEC in pT3. Larger samples and longer-term follow-up or prospective RCT are still needed to explore the indications of HIPEC in colorectal cancer. We have registered a prospective RCT (NCT04845490), including all cT4 patients received HIPEC, in order to further confirm the results of this study. We expect HIEPC to benefit patients with cT4, including those with unconfirmed pT4. In addition, we have already publish a paper (2) concerning the side effects in patients with pT3-pT4 colorectal cancer entitled “Safety evaluation of preventive intraperitoneal hyperthermic perfusion chemotherapy in patients with advanced colorectal cancer after radical resection”.
[1] Hompes D.; Tiek J.; Wolthuis A.; Fieuws S.; Penninckx F.; Van Cutsem E.; D'Hoore A. HIPEC in T4a colon cancer: a defendable treatment to improve oncologic outcome? Ann Oncol. 2012, 23, (12): 3123-9. DOI:10.1093/annonc/mds173
[2] Lin Yao, Guo Xikai, Shen Chu, Li Yuan, Wang Dianshi, Hu Peng, Wang Zheng, Wu Ke, Wu Chuanqing, Tao Kaixiong. Safety evaluation of preventive intraperitoneal hyperthermic perfusion chemotherapy in patients with advanced colorectal cancer after radical resection. Chin J Colorec Dis ( Electronic Edition ), 2021, 10(6): 591-598. DOI:10.3877/cma.j.issn.2095-3224.2021.06.005
Point 2: survival tables in figures 2 to 4. Title should be RFS survival non RFR (rate), since curves refer to survivals (as correctly reported in figure legends), non to rates. X-axis legend has misspelling (moths).
Response 2: We thank you for pointing this out. This is our mistake, thank you for the correction. We have fixed the figures.
We would like to express our great appreciation to you for comments on our paper. Looking forward to meet your approval.
Thank you and best regards.
Round 2
Reviewer 2 Report
The authors have adequately addressed my remarks.